# Effect of Heat Treatment on the Compressive Behavior of Zinc Alloy ZA27 Syntactic Foam

**DOI:** 10.3390/ma12050792

**Published:** 2019-03-07

**Authors:** Nima Movahedi, Graeme E. Murch, Irina V. Belova, Thomas Fiedler

**Affiliations:** Centre for Mass and Thermal Transport in Engineering Materials, School of Engineering, The University of Newcastle, Callaghan, NSW 2308, Australia; nima.movahedi@uon.edu.au (N.M.); graeme.murch@newcastle.edu.au (G.E.M.); irina.belova@newcastle.edu.au (I.V.B.)

**Keywords:** metallic syntactic foam, zinc alloy, expanded perlite, heat treatment, brittle and ductile deformation

## Abstract

Zinc alloy (ZA27) syntactic foams (SF) were manufactured using expanded perlite (EP) particles and counter-gravity infiltration casting. Due to a variation of the metallic matrix content, the density of the produced foam samples varied from 1.78 to 2.03 g·cm^−3^. As-cast and solution heat-treated samples were tested to investigate the compressive properties of the ZA27 syntactic foam. To this end, quasi-static compression tests were conducted. In addition, microstructural analysis of the as-cast and heat-treated syntactic foams was carried out using scanning electron microscopy. The results indicate that the heat treatment alters the microstructure of the ZA27 alloy matrix from a multiphase dendrite to a spheroidized microstructure with improved ductility. Moreover, the heat treatment considerably enhances the energy absorption and plateau stress $\left(\sigma_{\mathrm{pl}}\right)$ of the syntactic foam. Optical analysis of the syntactic foams under compression shows that the dominant deformation mechanism of the as-cast foams is brittle fracture. In comparison, the heat-treated samples undergo a more ductile deformation.

## 1. Introduction

In recent years, metallic foams have attracted a great deal of attention due to their unique mechanical and physical properties [1]. Their crashworthiness, energy absorption capacity and vibration damping make them promising candidate materials in industries such as automotive and construction [2,3,4]. Syntactic metal foams are a relatively novel subclass of metallic foams that use hollow particles embedded in a continuous metallic matrix. Syntactic foams are commonly considered as closed-cell foams, as a fluid is unable to permeate through the material [5]. The filler particles may differ, e.g., hollow sphere particles [6], fly ash cenospheres [7,8] and expanded volcanic particles [9] have been successfully used in previous studies. 

Up to now, most of the research has focused on the fabrication and evaluation of aluminum alloy syntactic foams. For example, Mondal et al. [10] produced an Al 2014 alloy syntactic foam using mono-sized cenospheres as the filler and studied the effect of particle volume fraction on the mechanical properties of the produced aluminum syntactic foam. Taherishargh et al. [11] fabricated Al syntactic foams by incorporating expanded perlite particles using different particle shapes within an A356 aluminum alloy matrix. Results indicate that near spherical particles result in improved mechanical properties compared to randomly-shaped perlite particles. 

Zinc and its alloys, most importantly the Zn-Al groups, are excellent engineering materials owing to their high damping capacity, good mechanical properties and corrosion resistance [12,13,14]. Moreover, the low melting point of Zn alloys and their good castability make them an attractive material choice for engineering applications in transportation and construction industries [15]. However, the application of Zn-based alloys has been restricted in industry due to their low ductility. To overcome this limitation, researchers have used heat treatment to enhance their mechanical properties. Venkatappa and Sharma investigated the solution treatment and ageing of various Zn-Al alloys [16]. Their results show that the thermal treatment enhanced the ductility and decreased the hardness of the tested alloys. Moreover, their study indicated that the heat treatment increased the homogeneity of the microstructure by dissolution of the dendritic phases via diffusion. Liu et al. [17] studied the effect of heat treatment on the mechanical properties of the ZA27 alloy used in this study. Ageing of the alloy (after solution heat treatment) at low temperatures was found to cause the precipitation of very fine phases that reduce the ductility of the alloy. On the other hand, ageing at high temperatures resulted in a coarse microstructure with a corresponding decreased tensile strength. According to their findings, ageing at a moderate temperature results in high ductility combined with intermediate hardness and tensile strength. 

The study of the structural and mechanical properties of Zn alloy foams has gained considerable momentum in recent years. Most of the published research has focused on Zn alloy foams produced with gas-releasing agent methods. Due to a non-uniform growth of gas bubbles during solidification, such foams contain pores with different shapes and sizes [18] that often follow a gravity-induced density gradient. As a result, this production technique is commonly associated with irregular porous structures and scattering mechanical properties. In contrast, the infiltration casting technique used in this study enables more uniform pore sizes, which are controlled by the selection of the embedded particles. Mondal et al. [19] fabricated a closed-cell ZA27 alloy composite foam with relative densities between 0.075 and 0.21. They mixed solid SiC particles (reinforcement) and CaH_2_ powder (foaming agent) with molten Zn alloy and investigated the quantitative effect of the foaming agent on the cell wall thickness and cell size of the obtained closed-cell foams. In comparison to unreinforced ZA22 foam, the addition of SiC particles embrittled the material and added large stress fluctuations to the compressive stress-strain curves. Furthermore, the compression of SiC reinforced ZA22 foam causes the formation of shear bands [20]. Kitazono et al. [21] used the powder metallurgy route to produce a closed-cell ZA22 foam with TiH_2_ powder as the foaming agent. Multiple ZA22 foam samples were produced using different holding times and heating temperatures resulting in a controlled variation of cell size and structure. 

Daoud produced syntactic foams using ZA22 alloy (with a solid density of 5.3 g·cm^−3^) by dispersion of Ni-coated fly ash microballoons in molten ZA22 using a stir casting method [22]. The densities of the produced syntactic foams were between 3.3 and 5.1 g·cm^−3^ for different volume fractions of the hollow particles. Shortcomings of stir casting are an inhomogeneous filler distribution due to buoyancy as well as breakage and subsequent melt infiltration of the hollow particles. These effects explain the relatively high density of these syntactic foams. The aim of the current research study is the manufacturing of Zn alloy syntactic foam with a significantly lower density than other research works. To this end, expanded perlite particles are combined with ZA27 alloy. The Perlite rock contains 2 to 6 vol.% water. The heating of this raw material above 870 °C increases its volume up to 15 to 20 times of its original volume. The expansion is due to a liquid to vapor phase transition of the trapped water in its structure [23,24]. The use of low-density volcanic particles permits the creation of ZA27 syntactic foam with a density range between 1.78 and 2.03 g·cm^−3^. To the authors’ knowledge, this is the lowest reported density for a Zn syntactic foam up to the present time. The mechanical behavior of the ZA27 syntactic foam and the effect of heat treatment are investigated using uni-axial compressive testing. 

## 2. Materials and Methods 

### 2.1. Sample Preparation

The alloy used for the metallic matrix of the manufactured syntactic foams was ZA27. According to ASTM B86-13 [25] this zinc alloy contains 27 wt.% aluminum and 2 wt.% copper as main alloying elements and its melting range is considered to be between 375 and 487 °C. The density and die shrinkage of this alloy are reported to be 5.00 g·cm^−3^ and 13.00 mm·m^−1^ respectively [26]. The size of expanded perlite (EP) particles was filtered using a vibrating sieve shaker. Particles trapped between meshes of spacing of 2 and 2.8 mm were collected for sample manufacturing. The EP particles were supplied by Australian Perlite Pty. According to the supplier information, the chemical composition of the EP particles is 75 wt.% SiO_2_, 14 wt.% Al2O_3_, 3 wt.% Na_2_O, 4 wt.% K_2_O, 1.3 wt.% CaO, 1 wt.% Fe_2_O_3_, 0.3 wt.% MgO, 0.2 wt.% TiO_2_ with traces of heavy metals [9]. Their particle and bulk densities are 0.16 g·cm^−3^ ($\rho_{\mathrm{EP}}$) and 0.091 g·cm^−3^ ($\rho_{B}$) [27]. Counter-gravity infiltration casting was used to manufacture cylindrical syntactic foam samples (see Figure 1) with a diameter of 20 ± 0.25 mm and 30 ± 0.2 mm height. This technique was also employed to manufacture aluminum alloy syntactic foams [28]. The method allows producing syntactic foam samples with similar densities and a uniform distribution of filler particles [9]. Therefore, it can be considered to be a repeatable production technique. In the first step, a stainless steel mesh was placed in the closed end of a graphite mold. Then, the mold was filled with four equally sized batches of expanded perlite particles. The resulting packed particle bed was compacted via vibration following the addition of each batch. To this end, the mold was gently tapped by hand for 20 s to ensure a homogeneous distribution of particles. The compaction aims to minimize the density gradient along the sample height. As a result, more homogenous material properties are achieved, and the overall foam density is decreased [9]. Another stainless steel mesh was placed on the open end of the mold to fix the particles in place. A solid block of ZA27 alloy was then inserted into a graphite crucible. The filled mold was inserted upside-down into the same crucible leaving only a tight gap to permit relative motion. In the next step, the assembly was moved inside a resistance furnace and heated to 535 °C (above the melting temperature of ZA27) for 30 min. After melting, the assembly was removed from the furnace and casting was immediately initiated by positioning a 1 kg weight on top of the mold, forcing it into the crucible. After cooling the samples under atmospheric conditions, they were manually removed from the mold. Finally, the stainless steel meshes were machined from the cast samples. 

The density of the produced foam was calculated by dividing the sample mass by its cylindrical volume. A near constant sample density was obtained (see Table 1) indicating that this is a repeatable process.

To investigate the effect of thermal treatment on the mechanical properties of ZA27 syntactic foam, heat treatment was performed on selected samples followed by water quenching and aging. The ZA27 alloy is solution heat-treated in the β (single-phase) region of the Zn-Al phase diagram. Higher temperatures during solution heat treatment accelerate the dissolution of α and η phases due to increased diffusion. According to [16] and [17] a supersaturated solution of β is achieved at 365 °C and the samples of the current study were kept at this temperature for 1 h before water quenching. Subsequent artificial ageing was performed in the α + η region at 140 °C for 24 h. This temperature was reported to yield the highest ductility for the ZA27 alloy [17]. 

In order to investigate the effect of heat treatment on the mechanical properties of the matrix alloy, four solid ZA27 samples with an aspect ratio similar to the syntactic foam samples (height 16 mm and diameter 10 mm) were tested. The samples were machined from the excess melt of the infiltration casting of the syntactic foam samples and thus solidified under similar conditions. Two of these samples underwent the same thermal treatment as the corresponding syntactic foam samples. 

### 2.2. Sample Characterization

Microstructural characterization of the samples was performed using a Zeiss Sigma VP FE-SEM Scanning Electron Microscope (Cambridge, UK) equipped with an energy dispersive spectroscopy (EDS, Bruker, Berlin, Germany) module and a BestScope microscope (BS-6010TTR, BestScope, Beijing, China) for optical microstructural study. The EDS line scan was performed to analyze the elemental distribution along the interface between matrix and particle to identify any possible chemical reaction between the ZA27 matrix and expanded perlite particles. EDS was also used to determine the chemical composition of the constituents. For the microscopy, sections were cut from the as-cast and heat-treated ZA27 syntactic foams and manually polished using silicon carbide grinding papers with grit sizes of 180, 240, 320, 600 and 1200. The final polishing was carried out with emery paper and 6 and 1 μm diamond suspension to achieve a mirror-like surface suitable for SEM observations. The polished samples were etched using Nital 2% solution for optical microstructural observations. X-ray diffraction technique using a Philips diffractometer (Philips Xpert, Almelo, The Netherlands) was employed to determine the phases in the microstructure of the ZA27 alloy. 

For geometrical characterization, the volume fraction of the ZA27 matrix ($F_{\mathrm{ZA}27}$), expanded perlite particles $\left(F_{\mathrm{EP}}\right)$ and voids $\left(F_{V}\right)$ within the syntactic foam structure were calculated using the following Equations [29]:(1)$$F_{\mathrm{ZA}27}=\frac{\frac{m_{\mathrm{SF}}-m_{\mathrm{EP}}}{\rho_{\mathrm{ZA}27}}}{V_{\mathrm{SF}}},$$
(2)$$m_{\mathrm{EP}}=\;\rho_{B\;}.\;V_{\mathrm{SF}},$$
(3)$$F_{EP}=\;\frac{\rho_{B}}{\rho_{P}},$$
(4)$$F_{V\;}=1-F_{\mathrm{ZA}27}-F_{\mathrm{EP}},$$where $m_{\mathrm{SF}}$ is the mass of the syntactic foam sample, $m_{\mathrm{EP}}$ is the (estimated) mass of contained EP particles, $V_{\mathrm{SF}}$ is the cylindrical sample volume, $\rho_{\mathrm{ZA}27}$ is the density of ZA27 alloy, $\rho_{B}$ is the EP bulk density, and $\rho_{\mathrm{EP}}$ is the EP envelope density. It must be highlighted here that the bulk and particles densities of expanded perlite beds were assumed to be constants according to [27] and as a result a constant fraction of $F_{EP}=\;\frac{\rho_{B}}{\rho_{P}}=56.88\%$ was obtained.

The syntactic foam (five as-cast and five heat-treated samples) and solid ZA27 samples (two as-cast and two heat–treated) were tested in quasi-static compression tests. All compression tests used a crosshead speed of 1 mm·min^−1^ on a 50 kN SHIMADZU uni-axial testing machine (AG-IS, SHIMADZU, Kyoto, Japan). Prior to testing, the cylindrical samples (solid and syntactic foams) were lubricated on both contact surfaces using CRC^®^ 5-56 (CRC Industries, NSW, Australia) multipurpose lubricant in order to minimize friction with the compression platens. The measured force-displacement data were evaluated following the ISO13314 standard [30]. Engineering strain ϵ and stress σ was obtained by division of the measured crosshead displacement and force with the initial sample length and cross section, respectively. A 5 kN SHIMADZU testing machine (AGS-X, SHIMADZU, Kyoto, Japan) was also used to determine the compressive behavior of expanded perlite particles. To this end, single EP particles with diameters of ~2.4 mm were compressed. 

Figure 2 shows a typical engineering stress-strain curve of a metallic foam and selected mechanical properties defined in the ISO13314 standard [30]. At small strains, a linear stress increase is visible which is followed by a stress peak. The slope of the linear section is the quasi-elastic gradient which is used to determine the 1% offset yield stress, i.e., the compressive stress at a plastic strain of 1%. Following the peak stress, the stress decreases towards a plateau region quantified by the plateau stress $\left(\sigma_{\mathrm{pl}}\right)$. This plateau stress ($\sigma_{\mathrm{pl}}$) is the arithmetic mean of the stress between 20% and 40% compressive strain. The plateau end stress is defined as 1.3 times the plateau stress and its corresponding strain is named the plateau end strain $\left(e_{\mathrm{ple}}\right)$. The energy absorption W of the syntactic foam samples was obtained by integration of the stress-strain data up to 50% strain:(5)$$W=\int_{0}^{0.5}\sigma \;d\varepsilon .$$

In addition, the energy absorption efficiency was determined using Equation (6):(6)$$\eta =\;\frac{W}{\sigma_{\max}\;.\;0.5}.$$

The ISO13314 standard [30] proposes the additional determination of an unloading modulus as a second elastic parameter that avoids measurement errors due to settling effects during the initial compression (albeit this elastic modulus corresponds to a slightly deformed foam). To this end, one as-cast and one heat-treated sample with the lowest density were compressed first. The obtained stress-strain data was used to define an unloading cycle for all subsequent samples between 70% (unloading) and 20% (reloading) of the determined plateau stress. 

## 3. Results and Discussion

### 3.1. Physical Properties 

The syntactic foam samples were divided into two groups, i.e., as-cast and heat-treated. Samples of similar density were assigned to different groups in an attempt to minimize its effect in this comparison (i.e., sample pairs with a similar density can be compared). Table 1 shows the physical properties of the syntactic foams. Their density varies from 1.78 to 2.03 g·cm^−3^. The inclusion of lightweight EP filler particles significantly reduces the foam density compared to the ZA27 matrix (5.00 g·cm^−3^). As a result, the obtained samples exhibit a distinctly lower density than other Zn-based syntactic foams [22]. Another explanation for this low density is the fact that EP particles cannot be significantly infiltrated with molten metal. Damage to the EP particle surface during casting will only expose a small fraction of internal pores to melt infiltration. This is different to e.g., cenospheres [31] which contain mostly a single large cavity that following particle fracture is completely infiltrated resulting in an increased foam density. 

As discussed above, the EP particle and bulk densities were considered to be constants. Based on this assumption, any deviation of the syntactic foam density is directly linked to the void fraction $F_{V}$. Table 1 shows that increasing the density decreases the void fraction. Voids are formed by casting defects, most importantly lack of melt infiltration and shrinkage cavities. 

### 3.2. Microstructural Observations 

Figure 3a shows an SEM image of the interface between the ZA27 matrix and an expanded perlite particle. In order to analyze the elemental distribution, an EDS line scan was performed along the shown arrow. The corresponding intensity profile in Figure 3b indicates silicon, oxygen and aluminum as the dominant elements along the expanded perlite section. The presence of aluminum in the expanded perlite section can be explained by Al_2_O_3_ being part of the chemical composition of the particles. As expected, mainly zinc and aluminum are detected within the ZA27 matrix. Figure 4 shows the XRD analysis of the as-cast solid ZA27 alloy. The XRD pattern of the as-cast matrix is shown in Figure 4. The (111) and (200) peaks of α, (0002), (10$\bar{1}$0) and (10$\bar{1}$1) of η, (0002) and (10$\bar{1}$1) of ε phases were detected. The intensity peaks for aluminum (see Figure 3) in the ZA27 matrix section likely indicate aluminum rich dendritic phases in the microstructure of the as-cast ZA27 alloy (see Figure 5). The EDS line scan further confirms the lack of any chemical reaction between the ceramic particles and the metallic matrix. The low intensity peaks of silicon and oxygen in the ZA27 matrix section are most likely due to perlite fragments that were deposited during sample polishing. Expanded perlite particle is composed of various oxides [9] that are unlikely to react with any constituents of the ZA27 alloy at a casting temperature of 535 °C. This result is in agreement with a previous study on syntactic foams containing EP particles within an A356 aluminum matrix [9]. 

Figure 5 shows backscattered electron images from the struts of as-cast and heat treated ZA27 syntactic foam. The microstructure of the struts in the as-cast foam is dendritic and composed of four different phases that have been identified using [17]. According to [17], the Al-rich α-phase appears dark and solidified first during casting (see arrow in Figure 5a). It forms the base of the dendrite structure and is surrounded by the Zn-rich β-phase, shown in grey color. In the final stages of the solidification, the Zn-rich ϵ- and η-phases (visible as bright areas) solidify in the inter-dendritic zones [17]. The dendritic structure of the α-phase changes due to the heat treatment (see arrow in Figure 5b). The thin and sharp α-phase observed in the as-cast condition transforms into more rounded grains due to dissolution of the dendritic phases in most parts of the heat-treated microstructure [16]. The higher magnification SEM images in Figure 5c,d show that the as-cast microstructure of the matrix is distinctly different from the heat-treated samples. The morphology of the interconnected α dendrites in as-cast condition transforms into a discontinuous and spheroidized morphology. In addition, the β phase in the as-cast condition exhibits a lamellar structure composed of α (dark), η and ϵ (light) layers [17]. SEM images of this phase before and after heat treatment reveal that the microstructure of the β-phase changes from fine lamellar (as-cast) to a rounded morphology (heat-treated) (see Figure 5e,f). This transition is known to increase the ductility of the ZA27 alloy [17]. 

Moreover, EDS analyses were performed to confirm the correct identification of each phase. To this end, EDS scans were conducted on points inside each phase to analyze its chemical composition. The obtained results (see Table 2) show the chemical composition of each phase in the as-cast sample. 

### 3.3. Mechanical Properties of Constituents

#### 3.3.1. Mechanical Properties of Solid ZA27 Samples

Two as-cast and heat-treated solid samples each were tested under uni-axial compression. The resulting engineering stress-strain curves are shown in Figure 6a. Brittle deformation was observed for the as-cast solid alloy with signs of macroscopic shear bands (dashed lines in Figure 6) at approximately 45° relative to the loading direction. In contrast, the heat-treated solid samples show a much more ductile deformation with pronounced barreling during compression (see arrows). Moreover, surface cracks are visible on the heat-treated sample, likely due to tensile stresses associated with the barreling. Thermal treatment decreased the proof stress from about 360 MPa (as-cast) to 280 MPa (heat-treated). At higher strains, a stress drop occurs in the as-cast samples. The stress drop commences at similar strains for both as-cast samples and coincides with the first occurrence of macroscopic shear bands. The findings of [17] could be confirmed, i.e., the applied heat treatment alters the deformation mechanism of the ZA 27 solid alloy from brittle to more ductile.

#### 3.3.2. Mechanical Properties of the EP Particles

The compressive force displacement curves of individual EP particles with diameters ~2.4 mm are shown in Figure 6b. Compared to the metallic matrix, the EP particles exhibit a very low strength which can be explained by their highly porous structure.

### 3.4. Mechanical Properties of ZA27 Syntactic Foam Samples

The compressive stress-strain curves of as-cast and heat-treated ZA27 syntactic foam samples are shown in Figure 7a,b. Similar to other porous materials, the curves exhibit three distinct regions. At small strains, a linear elastic response of the samples is apparent and is followed by a stress peak. Following this initial peak, the stress decreases towards a plateau region where the foam exhibits an approximately constant plateau stress $\left(\sigma_{\mathrm{pl}}\right)$. This behavior is very important for cellular materials when used as energy absorbers and a constant deformation resistance is targeted. At high strains, densification of the foam occurs. At this deformation stage, the porosity in the cellular structure has mostly disappeared due to pore (particle) collapse. As a result, the material behaves more like a solid metal and the stress increases sharply. 

Selected mechanical properties of ZA27 syntactic foam are shown in Figure 8 for both as-cast and heat-treated samples. The majority of mechanical properties increase with foam density. The exceptions to this trend are the energy absorption efficiency and the plateau end strain. A similar behavior was previously observed for aluminum-expanded perlite syntactic foams [32].

The unloading modulus (see Figure 8a) best characterizes the elastic stiffness of the syntactic foam. The lowest density samples in both groups (as-cast and heat-treated) were compressed without loading-unloading cycle according to the ISO 13314 standard (see Sample Characterization Section). A linear increase with the foam density is visible. The elastic unloading modulus increases from 3.3 to 4.2 GPa. As anticipated, this is a significant reduction in elastic stiffness compared to the Young’s modulus of solid ZA27 (77.9 GPa). 

The 1% proof stress of as-cast syntactic foam exceeds the corresponding value for heat-treated foam at a similar density. As described in Section 3.3, solid as-cast ZA27 has a higher yield stress compared to the heat-treated alloy. This behavior is now reflected in the syntactic foam properties as the Zn matrix supports most of the compressive load. The expanded perlite particles are too weak (see Figure 6b) to contribute significantly to the mechanical strength of the syntactic foam. The proof stress further increases significantly with foam density. A higher foam density corresponds to an increasing volume fraction of the load-carrying ZA27 matrix (see Table 1). In addition, the volume fraction of voids in the foam structure may influence the proof stress of the foams. The voids may trigger stress concentrations in the struts and thus decrease their load carrying capacity.

The plateau stress $\left(\sigma_{\mathrm{pl}}\right)$ is another important characteristic of metallic foams that quantifies their resistance to deformation during large-strain compression. Unlike the proof stress, the plateau stress of heat-treated samples exceeds the as-cast foams. The stress within the plateau region increases slightly with strain for all heat-treated samples (see Figure 7b). In the case of as-cast samples, distinct stress oscillations can be observed. These oscillations can be explained by the brittle deformation of the metallic matrix (see Discussion).

At a similar density, the heat-treated ZA27 syntactic foams exhibit a noticeably higher energy absorption compared to as-cast foams (Figure 8d). Increasing the density enhances the energy absorption of both cast and heat-treated foams. For both groups, the highest level of energy absorption (i.e., 16.90 and 30.40 MJ·m^−3^ for as-cast and heat-treated samples, respectively) coincides with the highest density. The improvement of energy absorption due to heat treatment can be explained by the ductility increase of the Zn alloy due to microstructural changes. 

The energy absorption efficiency of foam samples is shown in Figure 8e and appears to be largely independent of the foam density, i.e., it is near identical for all samples. In contrast, an apparently stochastic variation is found for the as-cast samples. These differences can be explained by changes in the macroscopic deformation mechanism from ductile to brittle deformation (see Discussion). In particular, sample AC-3 (density 1.93 g·cm^−3^) exhibits a significantly lower energy absorption efficiency compared to the other samples. This can be explained by the formation of a single large shear band that results in a significant stress drop up to high strains (see blue dash-dot line in Figure 7a).

The plateau end strain of as-cast ZA27 syntactic foams is distinctly higher compared to the heat-treated samples. This is further visible in the engineering stress-strain curves where the plateau region of the as-cast foam extends to higher strains (Figure 7). The plateau end strain is almost independent of foam density for the heat-treated foams (Figure 8f) and shows very little deviation for the as-cast samples. In summary, the plateau end strain can be considered vastly independent of the foam density. 

### 3.5. Failure Mechanisms

From points A to B, (see Figure 2) the samples compress primarily via elastic deformation. In the case of the as-cast samples, the stress increases gradually to a peak value (point B) before dropping towards a stress plateau due to shear band formation. Figure 9a,b shows the deformation of as-cast ZA27 syntactic foam with the maximum (2.03 g·cm^−3^) and minimum (1.78 g·cm^−3^) tested densities. In the lower density foam, the initial shear band forms at an angle of approximately 45° to the loading direction (see Figure 9b, ϵ=0.05). In the case of the higher density sample, the onset of macroscopic shear deformation occurs at a higher strain, i.e., ϵ≈0.1. However, only half of the sample surface is visible and shear bands may have formed earlier at the backside of the sample. An important characteristic of the compressive stress-strain curves of as-cast ZA27 syntactic foam samples is the stress oscillations in the plateau region. They can be explained by the brittleness of the as-cast matrix (see Figure 6a) and the resulting macroscopic deformation mechanism. Zn-based alloys usually have a hexagonal-close-packed (HCP) crystal structure with few slip systems for plastic deformation resulting in mostly brittle fracture. The brittle foam struts tend to break at relatively small local strains and the failure of a single strut increases the load on its neighbors thus eventually initiating a failure cascade. Heydari et al. [18] previously reported brittle fracture in metal foams with a similar ZA22 matrix during compression. Macroscopically, the consecutive strut fracture results in the formation of shear bands and a distinct plateau stress decrease. The subsequent increase of the plateau stress indicates that a shear band has been arrested. The stress then increases until the next shear band is initiated. This cyclic mechanism results in the distinct stress oscillations of the as-cast ZA27 syntactic foam. 

The deformation of heat-treated ZA27 syntactic foam is shown in Figure 9c,d. The stress-strain curves of the heat-treated foam samples are clearly different from the as-cast foam. Instead of stress oscillations, a smooth plateau region with gradually increasing stresses is found. Figure 6a indicates that the applied heat-treatment significantly enhances the ductility of solid ZA27 alloy, which can be explained by the microstructural changes of the struts. As described earlier, changing the morphology of the α phase from needle-like (as-cast) to spherical (heat-treated) enhances the ductility of the foam struts [16,17]. This microstructural change consequently alters the deformation mode of the ZA27 foam struts from brittle to ductile and the heat-treated foam struts are more likely to yield plastically than fracture. As a result, the compressive forces are more evenly distributed within the metallic matrix and shear bands are suppressed. In fact, no shear bands were observed in any of the heat-treated syntactic foam samples. Instead, the heat-treated ZA27 syntactic foam deforms by plastic yielding and buckling of struts. Plastic deformation commences in the weakest foam layer and the buckling of individual struts increases the load on their neighbors. Once the entire layer has collapsed, deformation shifts to the next weakest deformation band and so on. This results in a characteristic layer-by-layer collapse [33]. The ongoing plastic deformation (heat-treated foam) in comparison to brittle fracture and shear band formation (as-cast foam) requires more deformation energy and consequently results in an increased plateau stress and energy absorption of heat-treated syntactic foams. 

## 4. Conclusions

ZA27-expanded perlite syntactic foams were manufactured using counter-gravity infiltration casting. The effect of heat treatment on the ZA27 microstructure and its mechanical properties was investigated. Furthermore, the change in deformation behavior of heat-treated ZA27 syntactic foam was addressed. The key findings are:(1)ZA27 syntactic foam samples with low densities (1.78–2.03 g·cm^−3^) can be manufactured.(2)The applied heat treatment changes the microstructure of the ZA27 alloy from a dendritic structure to spherodized microstructure. This coincides with the results of a previous study [16,17](3)As previously observed in [17], the change of the microstructure increases the ductility of solid ZA27 alloy.(4)The increased matrix ductility changes the deformation mechanism of ZA27 syntactic foams from brittle fracture and macroscopic shear band formation to ductile deformation and layer-by-layer collapse.(5)The compressive stress-strain curves of as-cast ZA27 syntactic foam show distinct stress oscillations. This is linked to the propagation of macroscopic shear bands.(6)Heat-treated ZA27 syntactic foam exhibits a smooth stress plateau with gradually increasing stress. This is attributed to the ongoing plastic deformation and work hardening of foam struts.(7)Heat-treatment improves the specific energy absorption, plateau stress, and energy absorption efficiency of ZA 27 syntactic foam.

## Figures and Tables

**Figure 1 materials-12-00792-f001:**
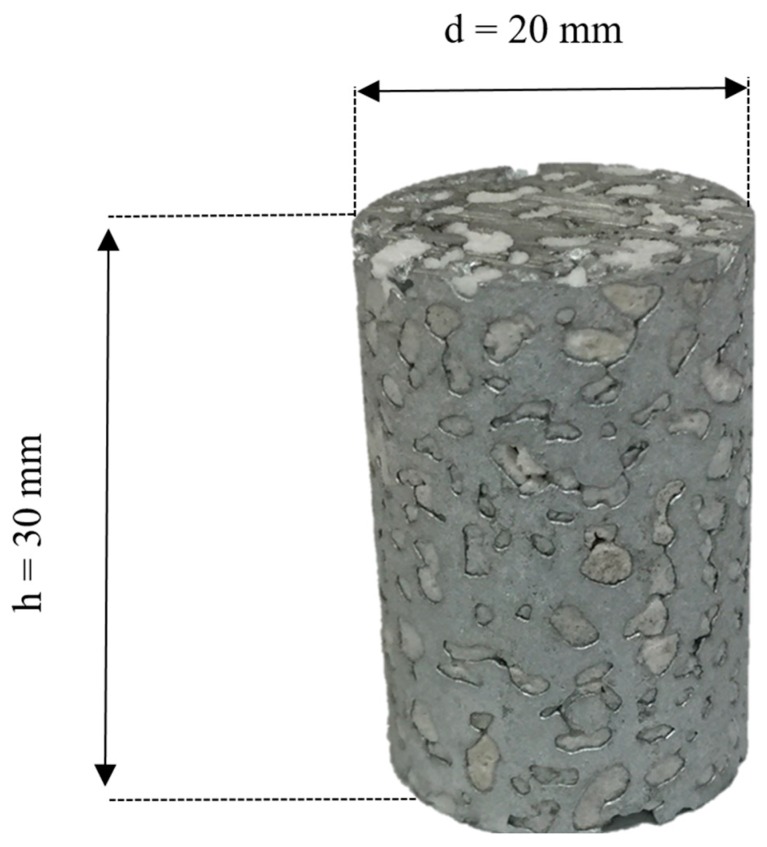
Cylindrical ZA27—Expanded perlite syntactic foam sample.

**Figure 2 materials-12-00792-f002:**
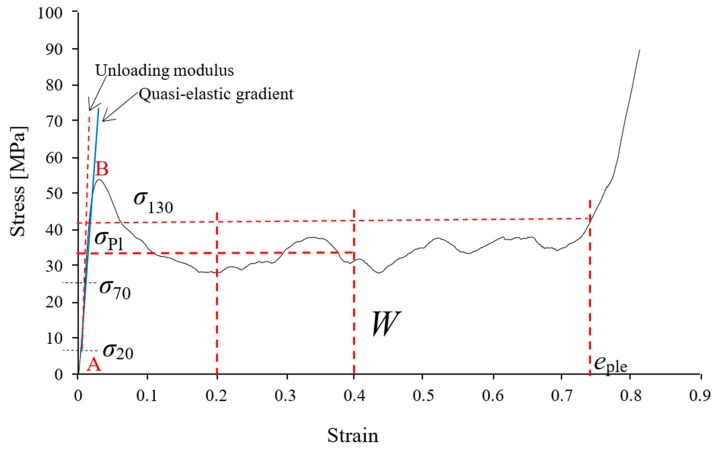
Engineering stress-strain curve of a metallic foam and related mechanical properties.

**Figure 3 materials-12-00792-f003:**
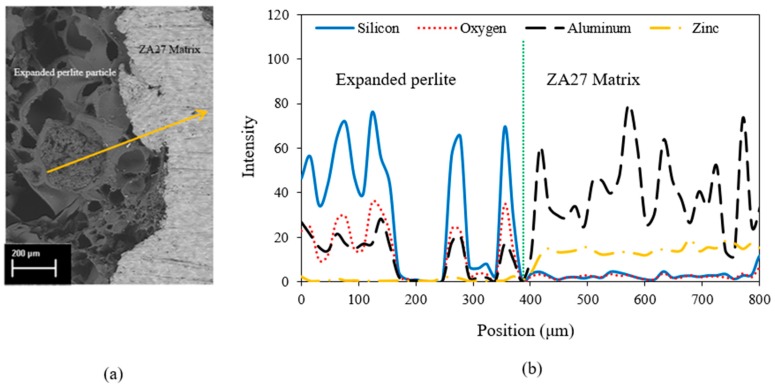
(**a**) SEM image of the interface between expanded perlite particle and ZA27 matrix. (**b**) Energy dispersive spectroscopy (EDS) line scan along the arrow in (a).

**Figure 4 materials-12-00792-f004:**
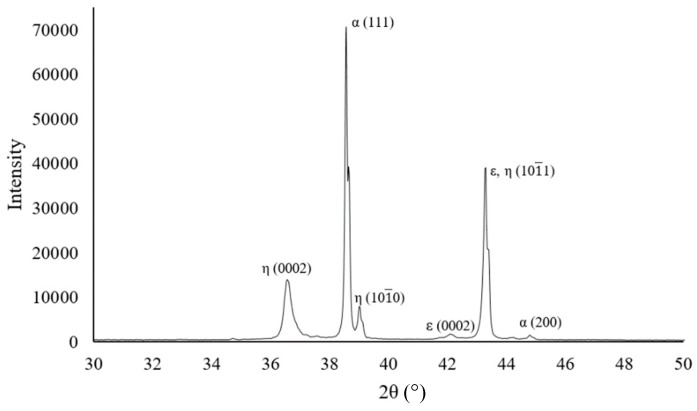
XRD pattern of as-cast solid ZA27 alloy.

**Figure 5 materials-12-00792-f005:**
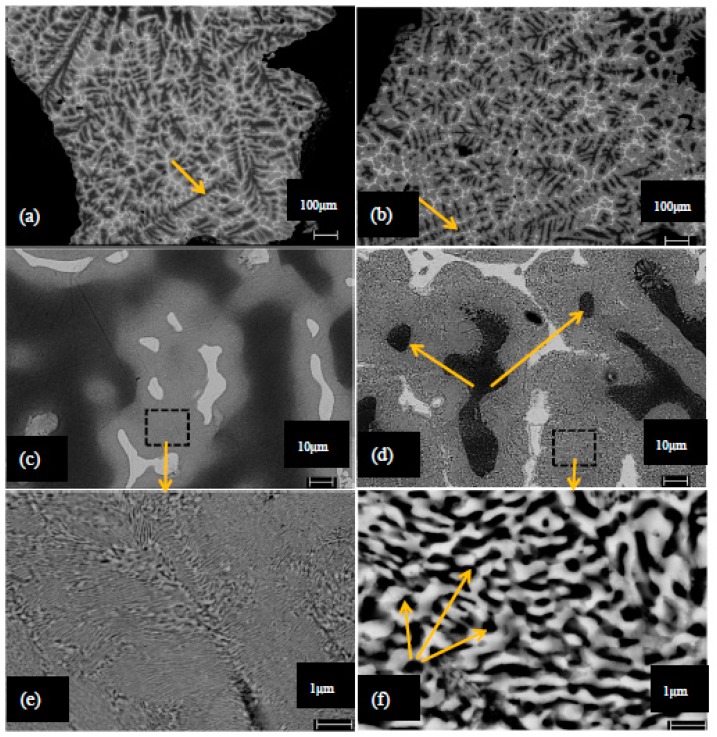
SEM images of cell walls in (**a**) as-cast and (**b**) heat-treated ZA27 syntactic foam. Magnified images of different phases in the cell walls in (**c**) as-cast and (**d**) heat-treated. Magnified SEM images of β phase in (**e**) as-cast and (**f**) heat-treated ZA27 syntactic foams.

**Figure 6 materials-12-00792-f006:**
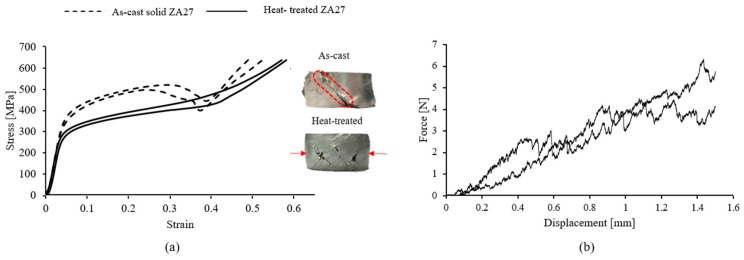
(**a**) Compressive stress-strain curves of as-cast and heat-treated ZA-27 solid samples. (**b**) Compressive force-displacement curves of individual expanded perlite (EP) particles.

**Figure 7 materials-12-00792-f007:**
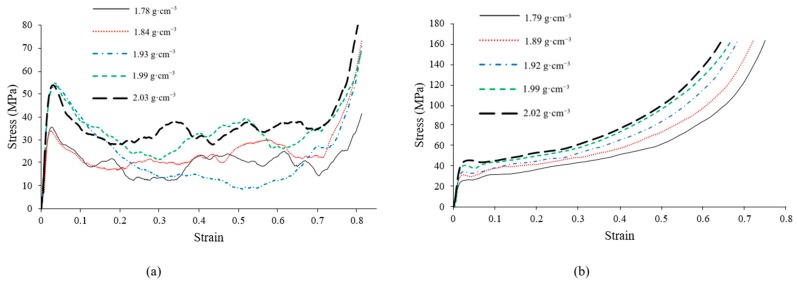
Compressive stress-strain curves of (**a**) as-cast and (**b**) heat-treated ZA27 syntactic foams with different densities.

**Figure 8 materials-12-00792-f008:**
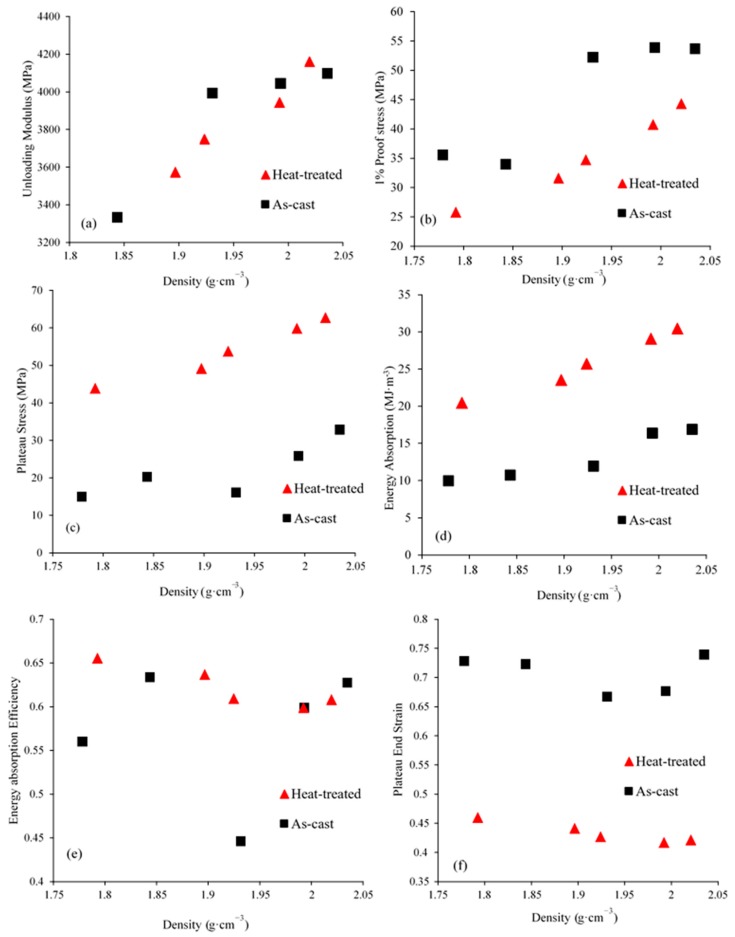
The principal mechanical properties of the as-cast and heat-treated ZA27 syntactic foam samples: (**a**) unloading modulus; (**b**) 1% proof stress; (**c**) plateau stress; (**d**) 50% strain energy absorption; (**e**) energy absorption efficiency and (**f**) plateau end strain versus sample density.

**Figure 9 materials-12-00792-f009:**
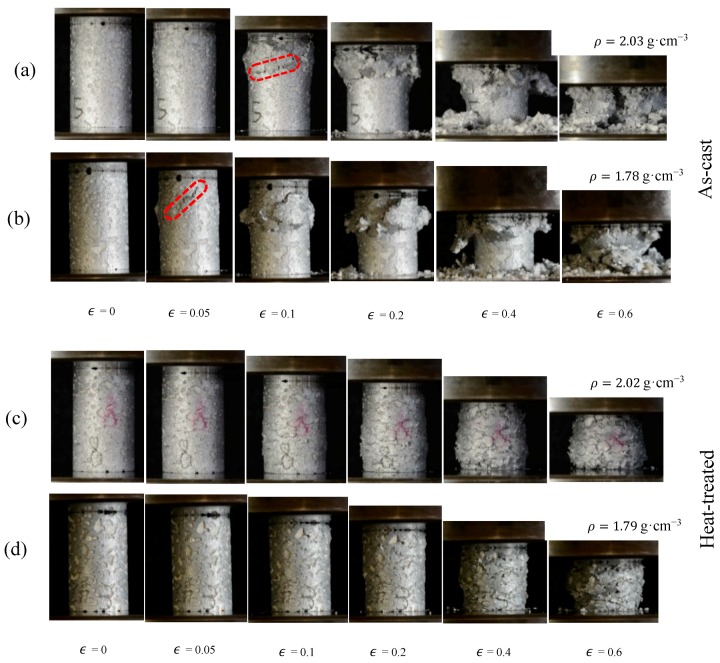
Deformation mechanism of (**a**,**b**) as-cast foams with densities of 2.03 and 1.78 g·cm^−3^ respectively. (**c**,**d**) heat-treated foams with densities of 2.02 and 1.79 g·cm^−3^ respectively.

**Table 1 materials-12-00792-t001:** Physical properties of the as-cast and heat-treated ZA27 syntactic foams.

Sample No*	SF Density (g·cm^−3^)	$F_{ZA27\;}(\%)$	$F_{EP}\;(\%)$	$F_{V}\;(\%)$
AC-1	1.78	33.75	56.88	9.37
AC-2	1.84	35.03	56.88	8.09
AC-3	1.93	36.80	56.88	6.32
AC-4	1.99	38.06	56.88	5.06
AC-5	2.03	38.89	56.88	4.23
HT-1	1.79	34.03	56.88	9.09
HT-2	1.89	36.12	56.88	7.00
HT-3	1.92	36.66	56.88	6.46
HT-4	1.99	38.04	56.88	5.08
HT-5	2.02	38.58	56.88	4.54

* AC: As-cast ZA27 syntactic foam, HT: Heat-treated ZA27 Syntactic foam.

**Table 2 materials-12-00792-t002:** EDS analysis of available phases in as-cast sample.

Phase	Zn (wt.%)	Al (wt.%)	Cu (wt.%)
α	44.07	55.15	0.78
ϵ	94.49	1.81	3.70
η	90.66	6.84	2.50

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
