# Peer review of "Effect of Heat Treatment on the Compressive Behavior of Zinc Alloy ZA27 Syntactic Foam"

_materials, 2019, doi:10.3390/ma12050792_

Round 1
Reviewer 1 Report
For a complete picture of the obtained results and a better understanding by the reader, it is recommended to perform XRD analysis to identify the phases in the material ZA 27 alloy and at the interface ZA 27 alloy expanded Perlite.
Presented the results obtained by EDS doesn’t give grounds for accurate classification and intermetallic compounds formed by heat treatment foam.
In the paper „…” the Authors already presented and defined this type of results for similar material.
It would also be good to provide information about Perlite expanding - for example, particle size analysis and in particular the chemical composition of the batch used. Perlite is a natural material and it is possible to change its composition depending on where the deposit comes from, and from which place exactly.
Do the authors think that changing the Perlite chemical composition may lead to interaction with Za27 Alloy ??
Author Response
Please find our reply in the attached work document.

Reviewer 2 Report
In this manuscript, the authors studied the compressive properties of Zinc AZ27 syntactic foam filled the expanded perlite particles with and without the heat treatment. In general, the reviewer agreed the methodology and results of this work, however, some terminologies need clarification:
1. The plateau stress was defined in the strain range of 20-40%. Why? In general, the plateau stress of foam structure materials is ideally used to express the plateau region at which the materials have the large deformation after the collapsed point. There, the void is filled during the compression. In reality, there is no perfect plateau region in the experimental stress-strain curve, there, it is normally quantified by averaging the plateau-like region up to the densification strain.
2. Can the plateau end strain be understood that it is also the densification one? If yes, it seems that the definition of this strain, at which the stress is reached 1.3 times the plateau stress, artificially affects to its result of the heat treatment sample.
3. In Figures 8c-d, it looks like the as-cast specimens were not collapsed uniformly. The top part seems to be failed first then the bottom one. The effect of the specimen size on the result is a question. The reviewer thinks the authors can verified by testing the specimen with lower ratio of length over diameter (maybe, 0.5 -1 is a good trial).
Other comments:
4. Please define the sample labels (ZFC and ZFH) in table 1.
5. Changing the line styles in Figure 3 is required for the print-out version.
6. Figures 4. g and h were shown but not presented.
7. In Figure 6, the line styles of the stress-strain curves for the lowest and highest density samples are same. It is difficult to follow with the print-out version.
Finally, this manuscript is recommended a major revision before publishing.
Author Response
Please find our replies in the attached word document.

Reviewer 3 Report
A list of suggestions and questions is the following:
1- QUESTION: ¿LOWER THAN? Explain it
Line 80 The aim of the current research study is the manufacturing of Zn alloy syntactic foam with a significantly lower density.
QUESTION: Report alloy melting point and shrinkage
89 The alloy used for the metallic matrix of the manufactured syntactic foams was ZA27. This zinc alloy contains 27wt% aluminum and 2wt% copper as main alloying elements
2-QUESTION: Report EP filler composition and compression strength. It is not technical to report line313: k (they can easily be crushed between one’s fingers)
Expanded perlite particles (EP filler) with a diameter range of 2-2.8 mm were selected as filler material.
QUESTION: It is not clear the EP envelope density, it is assumed a theoretical value: rep. REWITE: Their PARTICLE and bulk densities are 0.16 gNaN-3 (rEP) and 0.091 gNaN-3, (rB) according reference [23].
3-QUESTION: Report vibration conditions (time and frequency) to be sure that infiltration procedure is repeatable. Vibration density is not the same as bulk density and rb changes. Besides, vibration can break EP filler surface. Report EP filler distribution as a function of vibration conditions
QUESTION: Other authors have used Counter-gravity infiltration. Consider to include reference:
Deng-wei HUO, et al. Preparation of open-celled aluminum foams by counter-gravity infiltration casting. Transactions of Nonferrous Metals Society of China, 2012.
4-QUESTION: reference cast material properties such us density and shrinkage must be reported. Beside the next paragraph should be change to:
(line 120) four solid ZA27 samples with an aspect ratio similar to the syntactic foam samples (height 16 mm and diameter 10mm) were obtained from non-infiltrated zinc alloy thought machining. The samples solidified under similar conditions than syntactic foams. Two reference materials were heat treated in the same way than syntactic foam samples.
5-QUESTION: Explain if infiltration process is a repeatability process in case of uniform properties on infiltrated samples. Report if there are surface defects or non-uniform distribution of filler along the sample
6-QUESTION: It should be report the cooling curve of infiltration. Explain if the cooling rate on non-infiltrated zinc alloy is similar than syntactic foam. It is important to get data form conductivity properties
7-QUESTION: Arise line 130. EDS is well known.
8-QUESTION: Equation (1) is incomplete. If it is porcentual, add x100 and units (%)
9-QUESTION: Solve equation 3 Fep= 56,88 % since it was assumed data from ref 23
10-QUESTION:: Equation (4) is incomplete. If it is porcentual, add x100 and units (percentage)
11-QUESTION: Remove paragraph line 145 to 147.
… It assumes (Eq. 3) that the bulk and particle densities are not altered during the infiltration casting process
12-QUESTION: It is not clear the number of samples of syntactic foams mechanically tested.
(line 149) The syntactic foam and solid ZA27 samples were tested in quasi-static compression tests.
13-QUESTION: It should be included: From reference cast and machined AZ27 samples, two were tested as cast condition and two as heat treated condition.
14-QUESTION: Review references: line 153 ISO13314-11 standard [24], missed on line 171. On line 538 include: [24] ISO 13314-11 (YEAR), Mechanical Testing of Metals
15-QUESTION: Consider to change Figure 2 text to:
(line 155) Figure 2 shows the typical engineering stress-strain curve of a foam and selected mechanical properties following ISO13314-11 standard [24].
This figure it should not showed any results. It is experimental procedure. Besides, point A and B from figure 8a and 8b it must add. Both figures, Fig 8 a and b are recorded previously on fig 6, and they have to be removed
16-QUESTION: on line 1951, density of ZA27 is reported.¿ is the theoretical or measured.
17-QUESTION: Review reference data, for example on line 191: lower density than other Zn-based syntactic foams [22].
It is only ZA22 and density should be reported.
18-QUESTION: Remove paragraph line 197-199. Move to experimental procedure
19-QUESTION: Table 1 are not well explained ¿which material is cast. ¿ZFC-1?. Included cast or heat treated text into the table
20-QUESTION On this table, which is the faction of voids due to inner porosity of EP filler
21-QUESTION: References also report this result: (line 225): The EDS line scan further confirms the lack of any chemical reaction between the ceramic particles and the metallic matrix.
Include the reference.
22-QUESTION: I suggest reporting first results about Zn matrix (figure 4) instead Zn-Filler (figure 3)
23-QUESTION: On Figure 4. Remove pictures g and h. They are not commented on test and their information are reported on a and b.
24-QUESTION: Explain on figure 3, where… (line 265) Moreover, EDS analysis were performed to confirm the correct identification of each phase. To this end, EDS scans were conducted on points inside each phase
25-QUESTION: Consider to move line 272 to experimental procedure (figure 2) since this is a general behavior.
26-QUESTION: Structure of chapter 3.4 is not clear. Remove Italian letter and rewrite:
EXAMPLE: (REMOVE UNLOADING MODULUS) The unloading modulus (see Figure 7a) best characterizes the elastic stiffness of the syntactic foam. A linear increase with the foam density is visible
27QUESTION: Consider to move paragraph line 288 to 295 to experimental procedure (figure 2) since this is a general foam behavior.2
28 QUESTION: Make difference between conclusion and confirmation. Rewrite line 281
It can be concluded that the applied heat treatment alters the deformation mechanism of the ZA 27 alloy from brittle to more ductile.
TO: It was confirmed [17] that the applied heat treatment alters the deformation mechanism of the ZA 27 solid alloy from brittle to more ductile.
29-QUESTION: Remove line 313, as I suggested previously
30-QUESTION: Structure of chapter 3 and 4 should de change since discussion is reported on results on chapter 3 and some results are included on chapter 4 (discussion)
EXAMPLES: lines 334 , 335, 340
31-QUESTION: Improve data on Figure 7. Include tendency lines and legend on all graphs (not only on 7a)
32-QUESTION: Figure 7shows results on syntactic foams with densities from 1.75 to 2.05. However, on graph 7a, there is no data from foams with densities from 1.75 to 1.8. Justify this non-reported data.
33-QUESTION: Improve discussion of data from figure 7e, density 1.9-1.95 as cast. This value is far from the other data
34-QUESTION: Consider to modify structure of chapter 3 and 4 to: 3 Results and discussion and chapter 4 to 3.5 Compression failure mechanisms since chapter 4 should not include new data (figure 8)
35-QUESTION: Remove line 402 to 403 and rewrite to:
From points A to B (see Figure 2) the samples compress primarily via elastic deformation.
36- QUESTION: Conclusions included some confirmation data from other author’s results such us:
EXAMPLE: 2) The applied heat treatment changes the microstructure of the ZA27 alloy from a dendritic structure to spherodized microstructure… see reference 17
EXAMPLE: 3) The change of the microstructure increases the ductility of solid ZA27 alloy, see reference 17
Remove them or rewrite according with novel results.
Author Response

(The authors gave the same response as above.)

Round 2
Reviewer 2 Report
The reader agreed with the authors’ responses, and no more question was raised. The manuscript is recommended to plublish in the current version. Congratulations!
Reviewer 3 Report
Main recommendations have been included